# Recovery of IR700 Fluorescence After Near-Infrared Photoimmunotherapy: Discovery and Mechanistic Insights

**DOI:** 10.3390/cancers18010162

**Published:** 2026-01-02

**Authors:** Hideki Tanaka, Shuhei Okuyama, Ken Shirota, Mayumi Sugahara, Akiko Banba, Akihiro Ishikawa, Nobuhisa Minakata, Hirobumi Fuchigami, Masahiro Yasunaga, Tomonori Yano

**Affiliations:** 1Department of Otorhinolaryngology, Head and Neck Surgery, Tokyo Medical University, Tokyo 160-0023, Japan; hidekich@tokyo-med.ac.jp; 2Shimadzu Corporation, Kyoto 604-8511, Japan; okuyamas@shimadzu.co.jp (S.O.); shirota@shimadzu.co.jp (K.S.); nagata_a@shimadzu.co.jp (A.B.); ishikawa@shimadzu.co.jp (A.I.); 3Division of Developmental Therapeutics, Exploratory Oncology Research & Clinical Trial Center, National Cancer Center, Tokyo 104-0045, Japan; 4Department of Gastroenterology and Endoscopy, National Cancer Center Hospital East, Chiba 277-8577, Japan

**Keywords:** cancer, therapy, near-infrared photoimmunotherapy, tumor fluorescence, irradiation

## Abstract

This study investigates a novel phenomenon termed “early fluorescence recovery,” observed after near-infrared photoimmunotherapy (NIR-PIT) using IR700-conjugated antibodies. In mouse xenograft models treated with cetuximab-IR700, tumor fluorescence markedly decreased during NIR light irradiation but rapidly recovered within 10 min post-irradiation. This recovery was suppressed by L-sodium ascorbate, indicating the involvement of oxygen-dependent reactive processes, and it was accompanied by increased indocyanine green fluorescence, suggesting enhanced tumor perfusion. Therapeutically, divided irradiation administered after fluorescence recovery tended to achieve stronger tumor growth suppression than did single irradiation, although without statistical significance. These findings could demonstrate that early fluorescence recovery reflects transient reactivation of photoactive antibody–photoabsorber conjugates driven by molecular and vascular mechanisms; additionally, leveraging this brief recovery window may help optimize NIR-PIT treatment protocols.

## 1. Introduction

Near-infrared photoimmunotherapy (NIR-PIT) is a novel molecularly targeted therapy that combines the photosensitizer IRDye700DX (IR700) with antibody–photoabsorber conjugates (APCs) [1]. The therapeutic mechanism involves selective binding of APCs to tumor-associated antigens on the cancer cell membrane. Following irradiation with near-infrared (NIR) light, the hydrophilic side chains of IR700 dissociate, rapidly converting the APC into a hydrophobic form and altering its antibody structure [2,3]. These structural changes disrupt the tumor cell membrane, leading to rapid and extensive necrotic cell death. Because of this selective mechanism of action, NIR-PIT is considered a highly targeted and relatively safe cancer therapy. NIR-PIT can be applied to several tumor types simply by changing the IR700-conjugated antibody, and its application in various malignancies is being actively investigated [4,5,6,7,8]. In clinical trials on unresectable head and neck squamous cell carcinoma, NIR-PIT demonstrated promising efficacy and received accelerated approval in Japan in 2020 [9,10]. Recent real-world studies have demonstrated favorable disease control; however, adverse events such as pain and mucositis remain frequent [11]. In addition, post-treatment recurrence has been observed in some cases, indicating the need for further optimization of treatment methods [12,13]. IR700 exhibits fluorescence upon light excitation; however, this fluorescence diminishes following NIR irradiation due to structural changes in the molecule [14,15,16]. By leveraging this property, fluorescence imaging systems can be used to monitor IR700 signal dynamics, thereby quantifying the fraction of unreacted APCs within tumors. We previously demonstrated that intratumoral fluorescence decreases during NIR-PIT but does not decline to zero, instead reaching a plateau beyond a certain light dose. We suggested that this plateau reflects the threshold at which NIR-PIT achieves maximum therapeutic efficacy [17,18].

Additionally, we identified a novel phenomenon: intratumoral IR700 fluorescence intensity rapidly recovers immediately after the cessation of NIR irradiation. This immediate recovery differs from the enhanced permeability and retention effect, which is typically observed several hours after treatment, and represents an ultra-early response that begins within minutes of therapy completion [19,20].

Therefore, we aimed to describe the early post-treatment recovery of IR700 fluorescence, discuss its underlying mechanisms, and explore its potential implications for future optimization of NIR-PIT.

## 2. Materials and Methods

### 2.1. Cells and Cell Culture

In this study, we used A431 and FaDu-Luc2 cell lines obtained from the American Type Culture Collection (Manassas, VA, USA). The culturing method for A431 cells is described in a previous research paper [18]. As for FaDu-Luc2 cells, they were cultured in Eagle’s Minimum Essential Medium (EMEM; FUJIFILM Wako Pure Chemical Corporation, Osaka, Japan), supplemented with 10% FBS, 1% penicillin-streptomycin-amphotericin B suspension, and 8 μg/mL Blasticidin (InvivoGen, San Diego, CA, USA). The cultures were kept at 37 °C in 5% CO_2_ atmosphere.

### 2.2. Synthesis of IR700-Conjugated Antibodies

The cexuximab-IR700 conjugate used in this study was synthesized with cetuximab (Merck Biopharma, Tokyo, Japan), which is a human/mouse chimeric monoclonal antibody against the IgG1 subclass designed to target epidermal growth factor receptor (EGFR) and IRDye 700DX NHS ester (IR700; C_74_H_96_N_12_Na_4_O_27_S_6_Si_3_, molecular weight: 1954.22), which was kindly provided by Kansai Medical University (Appendix A). The synthesis procedure is described in detail in a previous paper [18]. After synthesis, an average of four IR700 molecules were found to bind to each antibody. We abbreviated cetuximab-IR700 as Cet-IR700.

### 2.3. Animal Model

In this study, six-week-old female BALB/c-nu/nu mice (The Jackson Laboratory, Yokohama, Japan) were employed. The mice underwent anesthesia with isoflurane and were inoculated with either 3.5 × 10^6^ A431 cells or 5 × 10^6^ FaDu-Luc2 cells, each suspended in 100 µL of phosphate-buffered saline (PBS), on the right or both dorsal sides. Tumor volume was determined using the formula TV = (L × W^2^)/2, where L and W represent the length and width of the tumor located beneath the skin, respectively, with TV indicating the tumor volume [21]. Tumor volume and weight were assessed at −1, 0, 1, 2, 4, and 7 days after NIR-PIT irradiation. Mice that attained a volume of 2000 mm^3^ or exhibited a weight loss greater than 20% were euthanized.

The animal experiments received approval from the Animal Experiment Committee at the National Cancer Center Japan. All procedures were conducted in accordance with the guidelines set forth by the Committee for the Care and Use of Laboratory Animals. These guidelines adhere to the ethical standards required by law and align with Japanese regulations regarding the use of laboratory animals.

### 2.4. In Vivo NIR-PIT

In vivo NIR-PIT experiments were performed using mice 6–7 days after inoculation with A431 or FaDu-Luc2 cells. Approximately 24 h after the intravenous administration of 100 μg Cet-IR700, the mice were subjected to in vivo NIR-PIT. The dose of Cet-IR700 was determined based on previous studies [17,18] and comparative results of therapeutic effects with dose changes (Appendix A). NIR light was irradiated by a 690 nm laser from a Shimadzu special laser module (peak wavelength; 690 ± 5 nm, Shimadzu Corporation, Kyoto, Japan) with a frontal diffuser (FD1, Medlight, Ecublens, Switzerland). Laser intensity was adjusted to 150 mW/cm^2^ and verified using an integrating sphere (S142C; Thorlabs Japan, Tokyo, Japan) connected to a power meter (PM100D; Thorlabs Japan). The anesthetized mice were placed at a position where the laser spot diameter was 15 mm on the tumor surface. The groups of mice used in each experiment are described below.

To observe fluorescence recovery after NIR-PIT, 10 mice with A431 tumors on the right flank at a volume of 200–460 mm^3^ were irradiated with NIR light at 50 J/cm^2^.

To observe fluorescence recovery with or without L-sodium ascorbate (L-NaAA), mice with A431 tumors on both flanks at a volume of 30–250 mm^3^ were used (n = 9). Approximately 24 h after intravenous Cet-IR700 administration, NIR light (50 J/cm^2^) was irradiated onto the tumor on the right flank, and the mice were subjected to fluorescence imaging. Next, the tumor was irradiated with NIR light on the left flank 15 min after the intraperitoneal administration of L-NaAA (80 mg in PBS 200 μL), and the mice were subjected to fluorescence imaging.

To perform divided light irradiation experiments, mice with A431 tumor volume of 40–90 mm^3^ were randomly divided into three groups as follows: (1) 100 μg of Cet-IR700, no NIR light (Control, n = 24); (2) 100 μg of Cet-IR700, NIR light at 50 J/cm^2^ (NIR-PIT, n = 15); and (3) 100 μg of Cet-IR700, NIR light at 25 J/cm^2^, 10 min interval, and 25 J/cm^2^ (Divided, n = 15).

To observe fluorescence recovery and perform divided light irradiation experiments in FaDu-Luc2 tumor model, mice with tumor volume of 40–90 mm^3^ were randomly assigned into the following three groups: (1) 100 μg of Cet-IR700, no NIR light (Control, n = 6); (2) 100 μg of Cet-IR700, NIR light at 50 J/cm^2^ (NIR-PIT, n = 7); and (3) 100 μg of Cet-IR700, NIR light at 25 J/cm^2^, 10 min interval, and 25 J/cm^2^ (Divided, n = 7).

### 2.5. In Vivo Fluorescence Imaging

We used an NIR fluorescence imaging system, LIGHTVISION (Shimadzu Corporation), to obtain visible and fluorescent images at a collection wavelength of ≥820 nm during, before, and after NIR-PIT (Appendix A). The LIGHTVISION camera head was placed approximately 50 cm above the observation surface.

Pre-NIR-PIT imaging was performed synchronously with a short-term laser irradiation of 250 ms using a Shimadzu special laser module. Using Shimadzu’s real-time fluorescence analysis software (Version 1.0.2), a region of interest (ROI) was set within the tumor region in the acquired visible image, and the fluorescence intensity was calculated in the same region.

During NIR light irradiation, both visible and fluorescent images were obtained at a rate of 10 frames per second, with the average pixel value in the ROI calculated in real-time for each fluorescent image. The average pixel value was normalized to the average pixel value in the ROI at the beginning of irradiation, and this result was referred to as the fluorescence intensity ratio, expressed in percentage units.

Fluorescence imaging after NIR-PIT was performed using the same method used before NIR-PIT at 5 min intervals until 30 min after NIR irradiation. The relative fluorescence intensities at 0 and 10 min after NIR light irradiation compared to the initial value were calculated, and fluorescence recovery was calculated using the following equation: fluorescence recovery [pt] = relative fluorescence at 10 min—relative fluorescence at 0 min.

### 2.6. In Vivo Bioluminescence Imaging (BLI)

Approximately 24 h after administration, BLI was performed before NIR irradiation. D-luciferin (Promega, Madison, WI, USA; 12.5 mg/mL, 200 μL) was administered intraperitoneally, and bioluminescence images were obtained using the IVIS Kinetic imaging system (PerkinElmer, Waltham, MA, USA) 10 min after administration. The acquired images were analyzed using the Living Image software (PerkinElmer). An ROI was placed in the tumor area of each acquired image, and the ROI signal was measured.

### 2.7. In Vivo ICG Imaging (Appendix A)

To assess blood flow, mice with A431 tumors on both flanks at a volume of 120–290 mm^3^ were used (n = 10). Fluorescence images were obtained before and after the intravenous administration of indocyanine green (ICG; 0.1 mM, 100 μL, Adooq Bioscience LLC, Irvine, CA, USA) with a 29-gauge needle and respiratory anesthesia. For ICG fluorescence imaging, LIGHTVISION (Shimadzu Corporation) was used; the excitation light was a 700–800 nm LED attached to the LIGHTVISION camera head with the same positioning as for NIR fluorescence imaging. Subsequently, only the tumor in the right flank was irradiated with 50 J/cm^2^ NIR light after ICG fluorescence imaging. The fluorescence intensity of each tumor was calculated relative to the non-tumor area between the two sides, and the ratio before and after NIR light irradiation was determined.

### 2.8. Fluorescence Microscopy

The procedure of fluorescence microscopy is described in the previous paper [18].

### 2.9. Histological Analysis

The procedure of hematoxylin and eosin (H&E) staining including sample preparation is described in the previous paper [18].

### 2.10. Statistical Analysis

Statistical analyses were conducted using EZR software (Version 4.3.1) [22]. For fluorescence recovery with or without L-NaAA and the fluorescence ratio, the Mann–Whitney U test was used. The Mann–Whitney U test was used to compare fluorescence recovery with or without L-NaAA, ICG fluorescence change with or without NIR light irradiation, and fluorescence recovery after NIR-PIT and divided light irradiation. The Steel–Dwass test was used for multiple comparisons of all groups. The error bars indicate the standard error. Unless otherwise specified, results are expressed as mean ± standard error. Statistical significance was set at *p* < 0.05.

## 3. Results

### 3.1. Fluorescence Intensity Recovered After NIR-PIT

A change in fluorescence was observed immediately after NIR-PIT (Figure 1A, Appendix A). As shown in Figure 1B, the fluorescence of A431 tumors was strong before NIR irradiation, decreased immediately after NIR irradiation, and partly recovered 10 min after irradiation. Fluorescence recovery was saturated at approximately 10 min, and no remarkable change was observed thereafter (Figure 1C).

### 3.2. Fluorescence Recovery Was Suppressed with L-NaAA

We hypothesized that fluorescence recovery is caused by multiple factors. First, the IR700 intermediate becomes excitable due to the production of reactive oxygen species; thereafter, it transitions to the ground state (Figure 2A). To clarify the mechanism of fluorescence recovery, we administered L-NaAA, which is known as an electron donor, to inhibit reactive oxygen species production and observed its effect on fluorescence recovery (Figure 2B,C, Appendix A). Fluorescence recovery was significantly suppressed in the absence of L-NaAA (*p* < 0.01 at 5, 10, 15, 20, 25, and 30 min after NIR light irradiation) [Figure 2D]. This finding suggests that fluorescence recovery results from the transition, thus supporting our hypothesis.

### 3.3. Blood Flow Increased After NIR-PIT

PIT increases blood flow in tumors (Figure 3A). We hypothesized that increased blood flow is a major factor that induces fluorescence recovery following NIR-PIT. To investigate this, we assessed the blood flow after NIR-PIT using ICG (Figure 3B). ICG fluorescence intensity was slightly decreased on the tumor without NIR-PIT (Control), whereas it was moderately increased on the tumor with NIR-PIT (*p* < 0.01) [Figure 3C,D]. These results suggest that blood flow increased shortly after NIR-PIT.

### 3.4. Divided Light Irradiation Showed a Tendency to Produce More Treatment Effect

To determine whether irradiation at the time of re-accumulation of the agents by fluorescence recovery was more effective, the antitumor effect of divided irradiation was compared with that of irradiation under clinical conditions using an A431 tumor mouse model (Figure 4A). Tumor growth in the NIR-PIT and Divided groups was significantly suppressed compared to that in the Control group (*p* < 0.05 in the NIR-PIT and Divided groups vs. Control group), whereas no significant difference was observed between the NIR-PIT and Divided groups (Figure 4B). In the Divided group, the relative tumor volume on day 7 tended to be more suppressed than that in the NIR-PIT group; however, there was no significant difference between the two groups (*p* < 0.01 at the NIR-PIT and Divided groups vs. Control group) (Figure 4C). No abnormal weight loss was observed in each groups (Appendix A).

Fluorescence recovery after NIR-PIT was determined (Figure 4D); the result showed no significant difference in the amount of fluorescence recovery after 50 J/cm^2^ irradiation between the NIR-PIT and Divided groups (Figure 4E, Appendix A).

Similar results were observed in FaDu-Luc2 head and neck squamous cancer cells (Appendix A). To compare antitumor effect of the divided NIR light irradiation, changes in bioluminescence signals indicating cellular functional activity were observed in FaDu-Luc2 cells. The signals in the NIR-PIT and Divided groups decreased significantly 1 day after NIR-PIT (Appendix A). Relative bioluminescence signal showed significant decrease in the NIR-PIT and Divided groups, whereas there was no significant difference in the Control group (*p* < 0.01 at the NIR-PIT and Divided group vs. Control group) (Appendix A). These results suggest that divided light irradiation could induce a greater therapeutic effect than NIR-PIT, although the difference was not significant.

## 4. Discussion

In this study, we identified a novel phenomenon of “early fluorescence recovery,” in which IR700 fluorescence, reduced immediately after NIR-PIT, rapidly recovered within a short time. We investigated the underlying mechanisms and explored their potential therapeutic implications. Our findings suggest that a divided irradiation strategy, in which irradiation is resumed after fluorescence recovery, may enhance tumor shrinkage.

NIR-PIT is a molecularly targeted therapy that uses APCs comprising IR700, an NIR photosensitizer, and an anti-EGFR antibody. First reported by Kobayashi et al. in 2011, NIR-PIT was a pioneering therapeutic approach [23]. In Japan, it has been approved for the treatment of unresectable locally advanced head and neck cancer and plays an important clinical role as a tumor-selective treatment with a favorable safety profile. Following NIR irradiation, IR700 undergoes structural modifications resulting in fluorescence loss [14,15]. This process involves radical anion formation and protonation, leading to cleavage of the intramolecular Si–O bond, hydrophilic-to-hydrophobic conversion, and subsequent aggregation. In vivo studies have shown that IR700 fluorescence in tumors decreases in proportion to the irradiation dose and plateaus beyond a certain threshold [17,18] even though Cet-IR700 stably accumulates in tumors and can emit fluorescence at least 24 h after administration [24]. However, fluorescence dynamics after irradiation have not been fully characterized. Using a non-cytotoxic low-energy pulsed irradiation method, we performed real-time monitoring and clarified the early fluorescence recovery phenomenon.

Early fluorescence recovery began immediately after irradiation and reached an equilibrium within approximately 10 min (Figure 1C). Two mechanisms were likely involved: (1) the transition of IR700 back to its ground state mediated by reactive oxygen species and (2) increased tumor blood flow via the super-enhanced permeability and retention (SUPR) effect. In NIR-PIT, the reaction of APCs with NIR light induces antibody aggregation, which causes plasma membrane damage and selective tumor cell death. This reaction was enhanced under hypoxic conditions or in the presence of ascorbic acid, where the triplet-state IR700 undergoes radical anion formation and ligand cleavage [25,26,27]. This results in irreversible fluorescence loss because the triplet excited state cannot be restored. In contrast, in oxygenated conditions, oxygen acts as an electron acceptor, producing singlet oxygen and enabling the return of IR700 to its ground state [27]. In the present study, sodium ascorbate administration significantly suppressed fluorescence recovery, supporting the hypothesis that early fluorescence recovery is closely related to the ground-state transition of IR700.

Additionally, tumors demonstrate enhanced angiogenesis, leading to the formation of immature vasculature under the effect of vascular endothelial growth factor. This phenomenon, whereby macromolecules accumulate preferentially within the tumor tissue, is known as the enhanced permeability and retention effect [28,29,30]. During NIR-PIT, irradiation induces rapid perivascular cell death, stromal expansion, and vascular dilation, which collectively increase tumor blood flow and promote molecular influx. This is referred to as the SUPR effect, and it is typically most evident within 6 h of irradiation and sustained for several hours thereafter [19]. In the present study, increased blood flow was observed immediately after irradiation, suggesting that early SUPR-mediated vascular changes contributed to fluorescence recovery. Furthermore, 24 h after administration of Cet-IR700, A431 tumors were excised before, immediately after, and 15 min after light irradiation, respectively, and subjected to histological analysis. Low fluorescence signal was observed immediately after light irradiation, whereas higher fluorescence was observed around blood vessels 15 min after light irradiation compared with other areas (Appendix A). If the fluorescence recovery is caused by the high fluorescence intensity near the blood vessel after light irradiation, this result suggests that the fluorescence recovery may have been affected by the SUPR effect.

Because early fluorescence recovery stabilizes within approximately 10 min, delivering a second irradiation after a brief waiting period may reactivate a larger pool of responsive IR700, thereby enhancing therapeutic efficacy. In this study, divided irradiation administered in two sessions yielded better tumor shrinkage than did a single irradiation, although the difference was not statistically significant. Previous studies have reported that repeated irradiation on subsequent days after initial treatment can produce strong antitumor effects [31,32]. However, because NIR-PIT requires general anesthesia, multi-day irradiation imposes a considerable clinical burden. Exploiting the short-term fluorescence recovery observed in the present study may enable effective split irradiation within a single anesthetic session, potentially reducing the barriers to clinical translation.

This study had several limitations. First, it was a preclinical study using in vivo models and may not fully reflect clinical responses. Second, no statistically significant difference in tumor shrinkage was observed between the split- and single-irradiation groups. This result may be explained by the limited sample size and the fact that the irradiation dose (50 J/cm^2^) was relatively high for the tumor size in the animal model. In our previous study, we demonstrated that approximately 20 J/cm^2^, corresponding to the plateau of IR700 fluorescence decay, represents the optimal dose for photoimmunotherapy [17,18]. Therefore, the lack of a difference observed in the present study may be attributable to the irradiation dose being set according to clinical standards, potentially leaving a small proportion of unreacted APCs even after a single irradiation. To address this issue, future studies with larger sample sizes are required to evaluate divided irradiation using appropriately optimized irradiation doses; for instance, terminating irradiation at the plateau of IR700 fluorescence decay is a potential optimization approach.

## 5. Conclusions

This study identified and characterized the phenomenon of early fluorescence recovery following NIR-PIT, in which IR700 fluorescence, diminished immediately after NIR irradiation, rapidly reappeared within minutes. Experimental analyses revealed that this recovery was mediated by the transition of IR700 to its ground state through reactive oxygen species and increased intratumoral blood flow shortly after treatment. These findings indicate that a brief recovery phase occurs in the tumor microenvironment immediately after irradiation, restoring a fraction of photo-reactive APCs. Exploiting this transient window through divided or sequential irradiation within a single session may enhance therapeutic efficacy while maintaining safety. However, further studies are warranted to validate this strategy in larger models and establish optimal irradiation parameters for clinical application.

## Figures and Tables

**Figure 1 cancers-18-00162-f001:**
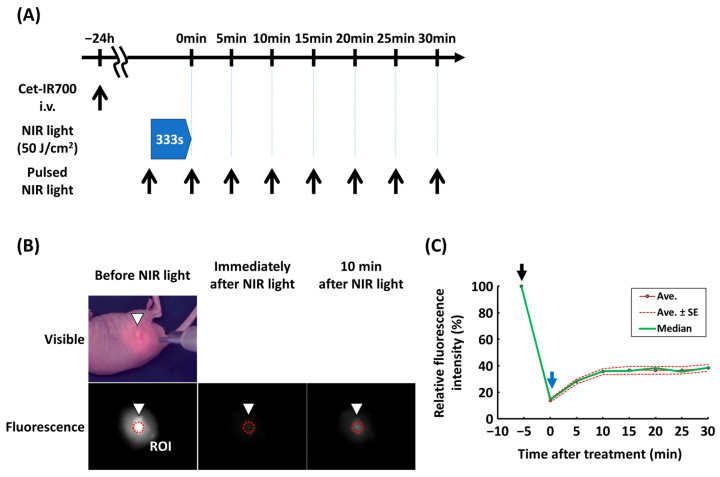
Fluorescence change during and after NIR-PIT in A431 tumor model in BALB/c nu/nu mouse. Fluorescence imaging on A431 tumor during and after EGFR-targeted NIR-PIT was performed in BALB/c mouse. (**A**) Schematic of the analysis of fluorescence recovery. (**B**) Representative visible and fluorescence images before NIR light, immediately after NIR light, and 10 min after NIR light. White triangle and red circle represent tumor bed and region of interest (ROI), respectively. (**C**) Relative fluorescence intensity when the fluorescence intensity is 100% before NIR light irradiation is 100%. Green, red, and dot-red curves indicate median, average, and average ± standard deviation, respectively (n = 10). Black and blue arrows indicate start and end of NIR light irradiation, respectively. NIR-PIT, near-infrared photoimmunotherapy.

**Figure 2 cancers-18-00162-f002:**
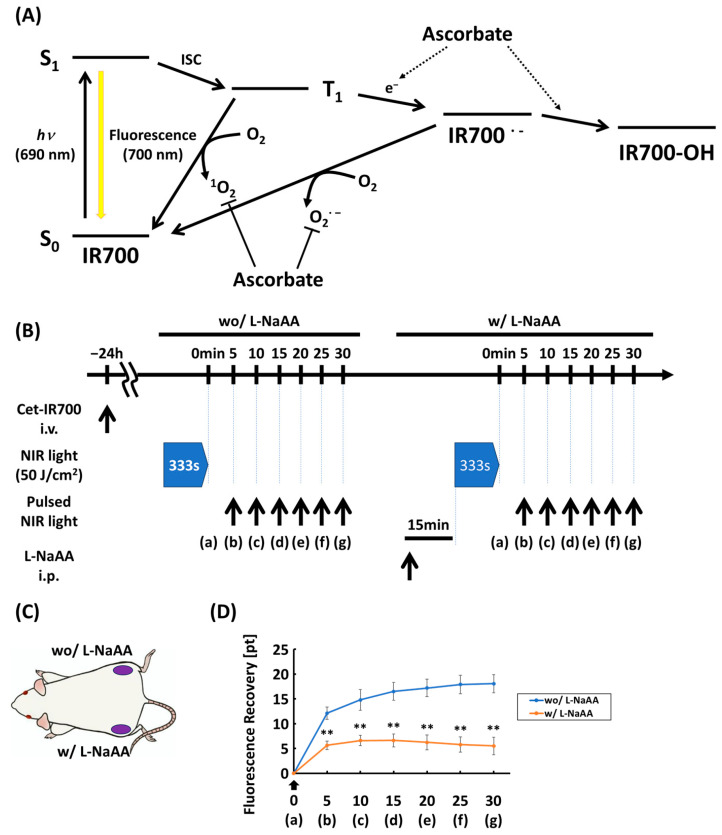
In vivo EGFR-targeted NIR-PIT with or without L-sodium ascorbate (L-NaAA). (**A**) Energy diagram and process of photo-induced ligand release reaction of IR700. (**B**,**C**) Schematic of the experiment. Fluorescence recovery without L-NaAA was observed on the tumor on the right flank; following the administration of L-NaAA, fluorescence recovery on the tumor on the left flank was observed. (**D**) Fluorescence recovery at 5 min intervals after NIR-PIT. Black arrow indicates end of NIR light irradiation (n = 9; Mann–Whitney U test; **, *p* < 0.01 in fluorescence recovery with L-NaAA vs. without L-NaAA); fluorescence recovery at 5 min with L-NaAA: 12.1 ± 3.7, without L-NaAA: 5.6 ± 2.5). NIR-PIT, near-infrared photoimmunotherapy.

**Figure 3 cancers-18-00162-f003:**
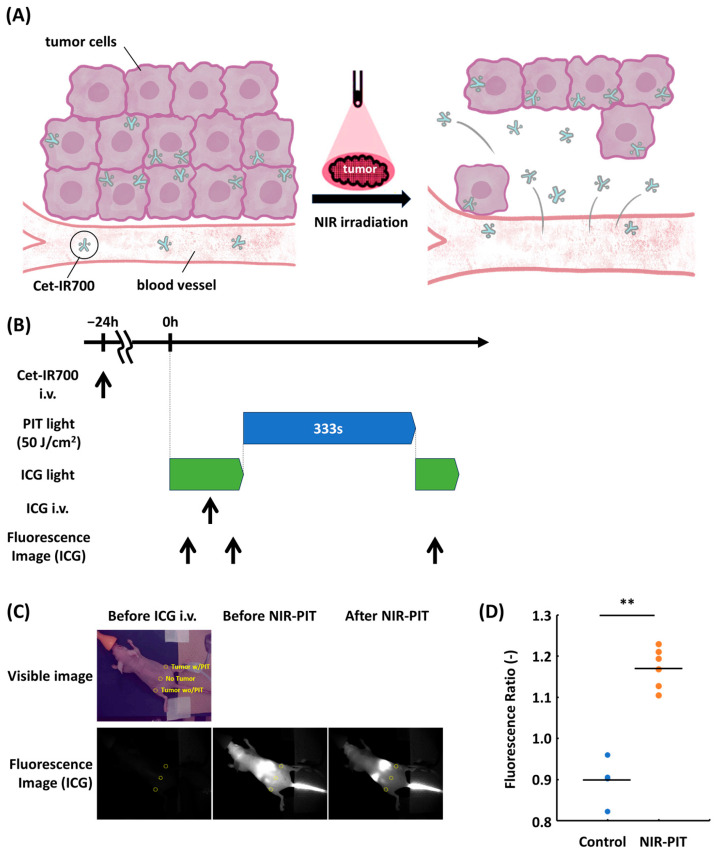
Assessment of blood flow with ICG following in vivo EGFR-targeted NIR-PIT. (**A**) Diagram of the super-enhanced permeability and retention effect. NIR-PIT induces stromal expansion and vascular dilation, thereby increasing blood flow and enhancing drug delivery. (**B**) Schematic of the experiment. (**C**) Representative visible and fluorescence images before ICG administration and before and after NIR-PIT. Yellow circles indicate ROI. (**D**) Fluorescence ratio of tumor with or without NIR-PIT compared to healthy tissue (n = 4 in the Control group, n = 6 in the NIR-PIT group, Mann–Whitney U test; **, *p* < 0.01). (Control; 0.90 ± 0.02, NIR-PIT; 1.17 ± 0.02). Black bar indicates average of each group (Control: 0.90, NIR-PIT: 1.17). ICG, indocyanine green; NIR-PIT, near-infrared photoimmunotherapy.

**Figure 4 cancers-18-00162-f004:**
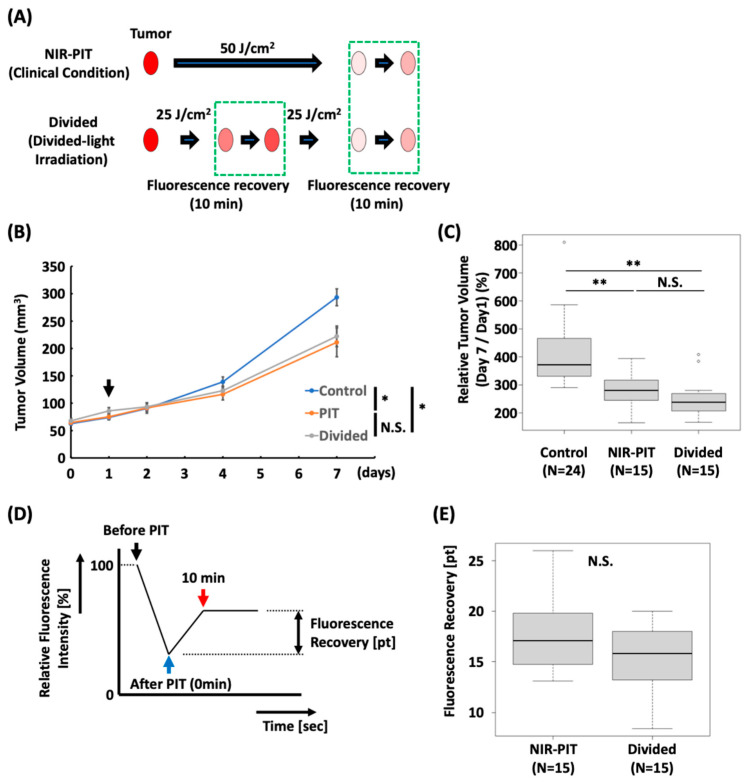
In vivo EGFR-targeted NIR-PIT with divided light irradiation in A431 tumor-mouse model. (**A**) Schematic of divided light irradiation. (**B**) Tumor growth curve. (n = 24 in the Control group, n = 15 in the NIR-PIT and Divided groups; Steel-Dwass test; *, *p* < 0.05 at day 7, N.S., not significant at day 7). Black arrow indicates the day of NIR light irradiation. (**C**) Relative tumor volume at day 7 compared to the volume at day 1 (n = 24 in the Control group, n = 15 in the NIR-PIT and Divided groups; Steel-Dwass test; **, *p* < 0.01 at day 7, N.S., not significant at day 7; Control: 409.6 ± 24.3, NIR-PIT: 283.0 ± 17.0, Divided: 251.4 ± 17.9). (**D**) Schematic of calculation of fluorescence recovery. (**E**) Fluorescence recovery 10 min after the end of NIR light irradiation. (n = 15; Mann–Whitney U test; N.S., not significant, NIR-PIT: 17.7 ± 0.96, Divided: 15.1 ± 0.9). NIR-PIT, near-infrared photoimmunotherapy.

## Data Availability

The data supporting the findings of this study are available from the corresponding author (Tomonori Yano) upon reasonable request.

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
