# Peer review of "Recovery of IR700 Fluorescence After Near-Infrared Photoimmunotherapy: Discovery and Mechanistic Insights"

_cancers, 2026, doi:10.3390/cancers18010162_

Round 1
Reviewer 1 Report
Comments and Suggestions for Authors
This is a highly intriguing paper that may contribute to proposing a new therapeutic strategy for photoimmunotherapy. However, one question that arises throughout the paper is whether fluorescence recovery can truly be interpreted as recovery of therapeutic efficacy, especially given that no significant difference in tumor size was observed. Although intratumoral fluorescence decreases during NIR-PIT, it does not fall to zero and instead reaches a plateau beyond a certain light dose. If the unreacted APCs come to be able to respond to a second light irradiation, this phenomenon may have therapeutic relevance. However, if the fluorescence recovery merely reflects APCs that have already reacted once reacting again, it may be unclear whether this would lead to any additional therapeutic benefit. Furthermore, although the authors state that fluorescence recovery was mediated by the transition of IR700 to its ground state, this conclusion appears insufficiently supported, and the evidence provided does not seem robust enough to definitively justify this interpretation.
Author Response
Reviewer 1 Comments and Suggestions for Authors:
This is a highly intriguing paper that may contribute to proposing a new therapeutic strategy for photoimmunotherapy. However, one question that arises throughout the paper is whether fluorescence recovery can truly be interpreted as recovery of therapeutic efficacy, especially given that no significant difference in tumor size was observed. Although intratumoral fluorescence decreases during NIR-PIT, it does not fall to zero and instead reaches a plateau beyond a certain light dose. If the unreacted APCs come to be able to respond to a second light irradiation, this phenomenon may have therapeutic relevance. However, if the fluorescence recovery merely reflects APCs that have already reacted once reacting again, it may be unclear whether this would lead to any additional therapeutic benefit. Furthermore, although the authors state that fluorescence recovery was mediated by the transition of IR700 to its ground state, this conclusion appears insufficiently supported, and the evidence provided does not seem robust enough to definitively justify this interpretation.
Response: We thank the reviewer for this important comment. As noted, no significant difference in therapeutic efficacy was observed between divided and standard irradiation, making it difficult to claim clinical superiority. We therefore agree that fluorescence recovery should not be directly interpreted as recovery of therapeutic efficacy. Accordingly, we have revised the conclusions to adopt more cautious and indirect language to better reflect this limitation, as described below.
The changes are highlighted in the manuscript.
p1 24-25
"These findings could demonstrate that early fluorescence recovery reflects transient reactivation of photoactive antibody–photoabsorber conjugates driven by molecular and vascular mechanisms"
p2 48-50
"Early fluorescence could recovery after NIR-PIT reflects transient reactivation of photoactive APCs through oxygen-dependent molecular and vascular mechanisms."
p9 281-282
"These results suggest that divided light irradiation could induces a greater therapeutic effect than does NIR-PIT, although the difference was not significant."
Reviewer 2 Report
Comments and Suggestions for Authors
This study examined the early recovery phenomenon of IR700 fluorescence following near-infrared photoimmunotherapy (NIR-PIT). This phenomenon is jointly driven by reactive oxygen species (ROS)-mediated transition of IR700 to its ground state and increased tumor blood flow, with explorations into its potential applications for optimizing treatment protocols. The article features a clear overall structure, rigorous logic, and originality, integrating mechanistic insights with therapeutic implications to offer valuable clinical guidance. The authors are advised to consider the following recommendations:
- Could the "early fluorescence recovery" phenomenon primarily result from partial quenching of IR700 fluorescence induced by near-infrared light irradiation of IR700?
- The authors are recommended to expand the discussion on potential confounding factors, such as tumor heterogeneity and IR700 photodegradation rate, and their influence on the "early fluorescence recovery" phenomenon.
- The dosage administered to mice was 100 mg of Cet-IR700 per animal. What is the rationale for this dosage? What is the solvent system used for 100 mg of Cet-IR700?
- What is the in vivo stability of Cet-IR700 in mice, and how is it metabolized? The authors are suggested to include a discussion on these aspects.
- The authors are recommended to incorporate relevant literature from 2024–2025 (e.g., updates on the latest advancements in NIR-PIT research).
Author Response
Comments and Suggestions for Authors: This study examined the early recovery phenomenon of IR700 fluorescence following near-infrared photoimmunotherapy (NIR-PIT). This phenomenon is jointly driven by reactive oxygen species (ROS)-mediated transition of IR700 to its ground state and increased tumor blood flow, with explorations into its potential applications for optimizing treatment protocols. The article features a clear overall structure, rigorous logic, and originality, integrating mechanistic insights with therapeutic implications to offer valuable clinical guidance. The authors are advised to consider the following recommendations:
Response: Thank you for your comment. I have changed manuscript as your advice. The changes are highlighted in the manuscript.
Comment1: Could the "early fluorescence recovery" phenomenon primarily result from partial quenching of IR700 fluorescence induced by near-infrared light irradiation of IR700?
Response1: With regard to the fluorescence recovery phenomenon, the key question may be whether it is attributable to (1) IR700 that was originally present on the tumor surface and partially quenched by irradiation, subsequently recovering its fluorescence, or (2) IR700 that was not initially present on the tumor but newly entered the tumor after light irradiation. We believe that the contributions of both mechanisms should be considered and cannot be disregarded.
Comment2: The authors are recommended to expand the discussion on potential confounding factors, such as tumor heterogeneity and IR700 photodegradation rate, and their influence on the "early fluorescence recovery" phenomenon.
Response2: I have added the following sentence in the discussion and supplementary figure S9.
p10 L338-p11 L345
“Furthermore, 24 hours after administration of Cet-IR700, A431 tumors were excised before, immediately after, and 15 minutes after light irradiation, respectively, and subjected to histological analysis. Low fluorescence signal was observed immediately after light irradiation, whereas higher fluorescence was observed around blood vessels 15 minutes after light irradiation compared with other areas (Figure S9). If the fluorescence recovery is caused by the high fluorescence intensity near the blood vessel after light irradiation, this result suggests that the fluorescence recovery may have been affected by the SUPR effect."
Comment3: The dosage administered to mice was 100 mg of Cet-IR700 per animal. What is the rationale for this dosage? What is the solvent system used for 100 mg of Cet-IR700?
Response3: I have added the following sentence in the Method and supplementary figure S9.
p3 L126-L128
“The dose of Cet-IR700 was determined based on previous studies [17,18] and comparative results of therapeutic effects with dose changes (Figure S10). "
Comment 4: What is the in vivo stability of Cet-IR700 in mice, and how is it metabolized? The authors are suggested to include a discussion on these aspects.
Response 4: I have added the following sentence in the Discussion.
p10 L310-L311
“even though Cet-IR700 stably accumulates in tumors and can emit fluorescence at least 24 hours after administration [24]. "
Comment5: The authors are recommended to incorporate relevant literature from 2024–2025 (e.g., updates on the latest advancements in NIR-PIT research).
Response5: I have added two new NIR-PIT research.
“8. Nagaya, T.; Nakamura, Y.; Okuyama, S.; Ogata, F.; Maruoka, Y.; Choyke, P.L.; Kobayashi, H. Near-infrared photoimmunotherapy targeting Nectin-4 in a preclinical model of bladder cancer. Cancer Lett. 2024, 585: 216606. https://doi.org/10.1016/j.canlet.2023.216606. “
“11. Okamoto, I.; Hasegawa, O.; Kushihashi, Y.; Masubuchi, T.; Tokashiki, K.; Tsukahara, K. Real-world effectiveness and safety of photoimmunotherapy for head and neck cancer: a multicenter retrospective study. Cancers (Basel) 2025, 17, 2671. https://doi.org/10.3390/cancers17162671."